# A Fluorescence-Based Method to Measure ADP/ATP Exchange of Recombinant Adenine Nucleotide Translocase in Liposomes

**DOI:** 10.3390/biom10050685

**Published:** 2020-04-29

**Authors:** Jürgen Kreiter, Eric Beitz, Elena E. Pohl

**Affiliations:** 1Institute of Physiology, Pathophysiology and Biophysics, Department of Biomedical Sciences, University of Veterinary Medicine Vienna, 1210 Vienna, Austria; 2Pharmaceutical and Medicinal Chemistry, Christian-Albrechts-University of Kiel, 24118 Kiel, Germany

**Keywords:** fluorescence, magnesium green^TM^ fluorescent dye, radioactivity, model systems, recombinant adenine nucleotide translocase, reconstitution into liposomes, mitochondria

## Abstract

Several mitochondrial proteins, such as adenine nucleotide translocase (ANT), aspartate/glutamate carrier, dicarboxylate carrier, and uncoupling proteins 2 and 3, are suggested to have dual transport functions. While the transport of charge (protons and anions) is characterized by an alteration in membrane conductance, investigating substrate transport is challenging. Currently, mainly radioactively labeled substrates are used, which are very expensive and require stringent precautions during their preparation and use. We present and evaluate a fluorescence-based method using Magnesium Green (MgGr^TM^), a Mg^2+^-sensitive dye suitable for measurement in liposomes. Given the different binding affinities of Mg^2+^ for ATP and ADP, changes in their concentrations can be detected. We obtained an ADP/ATP exchange rate of 3.49 ± 0.41 mmol/min/g of recombinant ANT1 reconstituted into unilamellar liposomes, which is comparable to values measured in mitochondria and proteoliposomes using a radioactivity assay. ADP/ATP exchange calculated from MgGr^TM^ fluorescence solely depends on the ANT1 content in liposomes and is inhibited by the ANT-specific inhibitors, bongkrekic acid and carboxyatractyloside. The application of MgGr^TM^ to investigate ADP/ATP exchange rates contributes to our understanding of ANT function in mitochondria and paves the way for the design of other substrate transport assays.

## 1. Introduction

In mitochondria, members of the solute carrier family 25 (SLC25) transport metabolically relevant substrates across the inner mitochondrial membrane [1,2,3,4,5,6]. The most prominent member is adenine nucleotide translocase (ANT, also AAC for ADP/ATP carrier protein), which exchanges ADP and ATP across the inner mitochondrial membrane to maintain energy balance in cells. The kinetics of ANT-mediated ADP/ATP exchange is mainly investigated using radioactive isotopes, such as ^3^H, ^14^C, or ^32^P. The utilization of radioactively labeled substances demands extensive precautions in the preparation of samples, conduct of measurements and waste disposal of radioactive material, which makes such experiments very tedious and limited to specially equipped laboratories.

A promising method to analyze ADP/ATP exchange in cells and mitochondria uses the fluorescent dye magnesium green (MgGr^TM^), which is sensitive to Mg^2+^ and benefits from the different binding affinities of Mg^2+^ to ATP and ADP [7]. However, the precision of the technique when used in cells is diminished by several limitations. The concentration of ATP and ADP in mitochondria is sensitive to the metabolic state and is significantly altered already under mild physiological and metabolic stress. Magnesium is the second most abundant cation in the cell and is involved in more than 300 cellular and mitochondrial enzymatic reactions, including metabolism of nucleic acids, lipids and proteins. In particular, Mg^2+^ participates in reactions involving the formation and use of ATP. Therefore, the assay specificity in living mitochondria is difficult to achieve and the estimation of the free Mg^2+^ concentration in mitochondria is delicate and may rapidly change during measurements [8]. The activity of another mitochondrial carrier, such as the ATP-Mg/Pi carrier, cannot be sorted out and may alter the kinetics of ANT-mediated ADP/ATP exchange in cells [9].

The application of recombinant ANT reconstituted into proteoliposomes allows to precisely predefine experimental conditions and the ADP/ATP exchange is directly dependent on the ANT activity. However, the model system is typically combined with the use of radioactively labeled substrates [10,11,12]. Although the fluorescence technique is more convenient, because it does not require radionucleotides, a protocol to measure ANT-mediated ADP/ATP exchange in proteoliposomes was not described, yet.

The goals of the study were: (i) to establish a fluorescence assay for measuring the ANT1 transport rate in proteoliposomes and compare to the radioactivity method; (ii) to show the correct transport function of the recombinant ANT1 produced in *Escherichia coli (E.coli)*; and (iii) to evaluate the activation and inhibition kinetics of ANT1-mediated ADP/ATP transport.

## 2. Materials and Methods

### 2.1. Chemicals

Sodium sulfate (Na_2_SO_4_), 2-(N-morpholino)ethanesulfonic acid (MES), tris(hydroxymethyl)- aminomethane (Tris), ethylene glycol-bis(β-aminoethyl ether)-N,N,N’,N’-tetraacetic acid (EGTA), bovine serum albumin (BSA), arachidonic acid (AA), magnesium dichloride (MgCl_2_), guanosine triphosphate (GTP), adenosine tri- and diphosphate (ATP, ADP), Sephadex™ G-50 size exclusion beads, bongkrekic acid (BA), carboxyatractyloside (CATR), DL-dithiothreitol (DTT), chloramphenicol, sodium lauryl sulfate, Triton^TM^ X 114, bromophenol blue and β-mercaptoethanol were purchased from Sigma–Aldrich GmbH (Neustadt, Germany). Sodium dihydrogen phosphate dihydrate (NaH_2_PO_4_) and EDTA were obtained from Merck (Darmstadt, Germany). 1,2-dioleoyl-sn-glycero-3-phosphocholine (DOPC), 1,2-dioleoyl-sn-glycero-3-phosphoethanolamine (DOPE) and cardiolipin (CL) were from Avanti Polar Lipids Inc. (Alabaster, AL, USA). Chloroform, sodium chloride (NaCl), tryptone/peptone ex casein, yeast extract, kanamycin sulfate, isopropyl β-D-thiogalactoside, ethylene glycol, Triton^TM^ X 100, glycerin and sodium dodecyl sulfate (SDS) were obtained from Carl Roth GmbH (Karlsruhe, Germany). N-Octylpolyoxyethylene came from BACHEM (Bubendorf, Switzerland). Magnesium Green^TM^, Pentapotassium salt, cell impermeant (MgGr^TM^) was purchased from Thermo Fisher Scientific (Waltham, MA, USA) and [2,5′,8-^3^H] ATP from Perkin Elmer (A = 1 mCi; Waltham, MA, USA).

### 2.2. Cloning, Isolation and Reconstitution of Murine ANT1 and UCP1

Murine ANT1 (Slc25a4)-containing proteoliposomes were produced according to a previously established protocol for mUCP1 [13]. The open reading frame of ANT1 was obtained as a cDNA clone (IRAVp968G119D; BioCat, Heidelberg, Germany) and inserted between *Nde*I and *Bam*HI restriction sites of the expression vector pET24a (Novagen, Darmstadt, Germany). Expression plasmids were transformed into the *E. coli* expression strain Rosetta (DE3; Novagen). Bacteria were grown in DYT-media containing 16 mg/mL peptone ex casein, 10 mg/mL yeast extract, 5 mg/mL NaCl, 25 µg/mL kanamycin sulfate and 34 µg/mL chloramphenicol until an optical density OD_600_ of 0.5 was reached. Protein expression was induced using 1 mM isopropyl β-D-thiogalactoside. Bacteria were harvested after 3 h. To isolate inclusion bodies, bacterial pellets were re-suspended in 100 mM Tris, 5 mM EDTA, pH 7.5 (TE–buffer) containing 1 mM DTT and Protease Inhibitor Cocktail for bacterial extracts (Sigma–Aldrich, Vienna, Austria) and disrupted by applying a high-pressure homogenizer One Shot (Constant Systems Limited, Daventry, UK) at 1 kbar. The cell lysate was centrifuged for 30 min at 15,000× *g* and the pellet was re-suspended in 150 mM NaH_2_PO_4_ at pH 7.9, 25 mM EDTA, 5% ethylene glycol (PA-buffer) plus 2% Triton X-100, 1 mM DTT and protease inhibitor. Inclusion bodies were obtained after centrifugation at 14,000× *g* for 20 min.

For protein reconstitution, 1 mg protein from inclusion bodies was solubilized in 100 mM Tris at pH 7.5, 5 mM EDTA, 10% glycerin (TE/G-buffer) containing 2% sodium lauryl sulfate and 1 mM DTT, and mixed gradually with 50 mg lipid mixture (DOPC, DOPE and CL; 45:45:10 mol%) dissolved in TE/G-buffer plus 1.3% Triton X-114, 0.3% n-octylpolyoxyethylene, 1 mM DTT and GTP to a final concentration of 2 mM. After 3 h of incubation, the mixture was concentrated to a fifth using Amicon Ultra-15 filters (Millipore, Schwalbach, Germany), dialyzed for 2 h against TE/G buffer with 1 mg/mL BSA and 1 mM DTT, and then two times without DTT in a total time of at least 12 h. The mixture was dialyzed three times against assay buffer (50 mM Na_2_SO_4_, 10 mM MES, 10 mM Tris, 0.6 mM EGTA at pH 7.35) for buffer exchange. To eliminate aggregated and unfolded proteins, the dialysate was centrifuged at 14,000 × g for 10 min and run through a 0.5 g hydroxyapatite-containing column (Bio-Rad, Munich, Germany). Non-ionic detergents were removed by application of Bio-Beads SM-2 (Bio-Rad). Proteoliposomes were stored at −80 °C. The protein concentration in proteoliposomes was measured using a Micro BCA^TM^ Protein Assay Kit (Thermo Fisher Scientific, Prod. #23235, Waltham, MA, USA). Protein purity was verified by SDS–polyacrylamide gel electrophoresis (PAGE) plus silver staining.

Production, purification, and reconstitution of recombinant murine UCP1 into proteoliposomes followed a previously published protocol [14].

### 2.3. SDS–PAGE and Silver Staining

For SDS–PAGE, approximately 0.5 µg of inclusion body proteins solubilized in 1% SDS or proteoliposomes was mixed with loading buffer containing bromophenol blue to a concentration of 0.025 M Tris pH 6.0, 2.5 % glycerin, 1% SDS, and 1% β-mercaptoethanol, and degraded at 97 °C for 10 min. Samples and Precision Plus Protein Dual Color Standards (Bio-Rad, Vienna, Austria) were loaded on 15% SDS–PAGE gels and electrophoresis was performed at 80 V for 30 min for at least 2 h at 120 V. Silver staining of the gel was performed according to [15]. The purity of recombinant ANT1 and UCP1 is shown in Appendix A.

### 2.4. Preparation of Unilamellar (Proteo-) Liposomes

DOPC, DOPE and CL lipids were mixed in chloroform at 45:45:10 mol%, respectively, and evaporated under nitrogen flow until they assembled as a thin film on the wall of a glass vial. Buffer containing 50 mM Na_2_SO_4_, 10 mM Tris, 10 mM MES, and 0.6 mM EGTA at pH = 7.34 was added to the lipids and the solution vortexed until the lipids were fully dissolved. Liposomes and ANT1- or UCP1-containing proteoliposomes were then diluted to a final lipid concentration of 1 mg/mL. Unilamellar (proteo-) liposomes were formed by a Mini-Extruder system (Avanti Polar Lipids Inc., Alabaster, AL, USA) using a membrane filter with a pore diameter of 100 nm (Appendix A).

### 2.5. Calibration of Fluorescence Intensity of MgGr^TM^

For calibration, the fluorescence intensity of 3 µM MgGr^TM^ fluorescent dye was measured at Mg^2+^ concentrations from 0 to 1.2 mM in 0.2 mM increments in buffer solution. The binding constant of Mg^2+^ to MgGr^TM^ was estimated by the fit of an exponential function to the data [7]. The fluorescence signal of 3 µM MgGr^TM^ in the presence of 1 mM Mg^2+^ and 0, 0.25, 0.5, 1, 2, and 3 mM ATP or ADP was first converted to a free Mg^2+^ concentration in buffer. Binding constants of ADP/ATP to Mg^2+^ were determined by fitting a Michaelis–Menten equation to the data [7]. All measurements were performed in a 96-well plate (TPP, Trasadingen, Switzerland) with V_Well_ = 300 µL. Fluorescence intensity was measured as counts per second in a plate reader (EnSpire^®^ Multimode Plate Reader, Perkin Elmer, Waltham, MA, USA) with excitation and detection wavelengths of 506 and 531 nm, respectively.

### 2.6. Fluorescence-Based ADP/ATP Exchange Rate Measurements

Prior to extrusion, (proteo-) liposomes were incubated with 3 µM MgGr^TM^, 1 mM MgCl_2_ and 2 mM ATP for at least 20 min at T = 4 °C. After extrusion, extraliposomal substrates were removed by size-exclusion chromatography. The inhibitors CATR and BA were added before extrusion and after size-exclusion at the concentrations indicated in the figure descriptions to account for the random orientation of proteins in liposomes. Samples were added to a 96-well plate and the time course of fluorescence intensity was recorded in a plate reader. After the signal stabilized, ADP/ATP exchange was initiated by the addition of 2 mM ADP to the buffer solution. The concentration of ATP inside the (proteo-) liposomes was calculated from the fluorescence signal using the measured binding constants of Mg^2+^ to MgGr^TM^, ATP, and ADP.

### 2.7. Radioactivity-Based Exchange Rate Measurements

Prior to extrusion, 2 mM ADP was added to (proteo-)liposomes. After extrusion, external ADP was removed by size-exclusion chromatography. ADP/ATP exchange was initiated by the addition of 2 mM ^3^H labeled-ATP and stopped after size exclusion chromatography. Radioactivity was measured by liquid scintillation counting (Tri-Carb 2100TR, Perkin Elmer). For inhibition, 100 µM of each of the ANT-specific inhibitors, CATR and BA, were simultaneously added prior to ^3^H-ATP to account for the random orientation of ANT1 in the membrane.

### 2.8. Statistical Analysis

Data analysis and fitting were performed by Sigma Plot 12.5 (Systat Software GmbH, Erkrath, Germany) and are displayed as the mean ± standard deviation (SD) of at least three independent measurements.

## 3. Results

### 3.1. Calibration of MgGr^TM^ Fluorescence Intensity to ADP/ATP Concentration and Determination of Related Binding Affinities

We used the fluorescent dye MgGr^TM^, whose emission efficiency significantly increases in the presence of free Mg^2+^ (Figure 1A). Since Mg^2+^ possesses different binding affinities for MgGr^TM^, ATP, and ADP, it serves as a mediator between fluorescence intensity and ATP concentration. To convert fluorescence intensity to purine nucleotide (PN) concentration, the binding affinities of Mg^2+^ for MgGr^TM^, ATP, and ADP were measured: K_Mg,MgGr_ = 1.18 ± 0.11 mM (Figure 1B), K_Mg,ATP_ = 0.44 ± 0.09 mM (Figure 1C) and K_Mg,ADP_ = 2.23 ± 0.50 mM (Figure 1C).

### 3.2. Conversion of MgGr^TM^ Fluorescence Intensity to ADP/ATP Exchange Rate Values

(Proteo-)liposomes were filled with MgGr^TM^, Mg^2+^ and ATP, and external substrates were removed by size exclusion chromatography (Figure 2A, left image). The addition of ADP initiated ATP exchange with ADP through ANT1. This led to Mg^2+^ binding to MgGr^TM^ (Figure 2A, right image) increasing fluorescence intensity with time (Figure 2B).

The Mg^2+^ concentration inside proteoliposomes was directly determined using previously determined binding affinities (Figure 2C). The time course of the ATP decrease inside proteoliposomes was calculated based on the assumption that the ratio of ANT1-mediated ADP/ATP exchange was 1:1 (Figure 2D). From an exponential fit of the data, we calculated the exchange rates of k_none_ = 7.8 ± 0.6 µM/min, k_ANT1_ = 35.3 ± 3.2 µM/min and k_ANT1+BA/CATR_ = 14.7 ± 1.0 µM/min (Figure 2E).

### 3.3. Measurements of ADP/ATP Exchange Using Radioactivity Assay

To evaluate the MgGr^TM^ fluorescence assay, we performed radioactivity measurements with ^3^H-labeled ATP. We measured the time course of ^3^H-ATP uptake into (proteo-) liposomes, which were initially filled with ADP (Figure 3A, left image). By the addition of ^3^H-ATP, ANT1-mediated ADP/ATP exchange was initiated (Figure 3A, right image).

After 1, 20 and 60 min, the (proteo-) liposome containing solution was added to a size exclusion chromatography column and extra-liposomal ^3^H-ATP was removed. The flow-through was collected, a scintillation cocktail added, and the amount of ^3^H-ATP determined by liquid scintillation counting (Figure 3B). From the fit of an exponential function to the data sets, we calculated initial ATP uptake rates of k_none_ = 2.6 ± 0.9 µM/min, k_ANT1_ = 28.4 ± 10.9 µM/min, and k_ANT1+BA/CATR_ = 7.2 ± 2.1 µM/min (Figure 3C).

### 3.4. Proof of Specificity of ANT1-Mediated ADP/ATP Exchange Rate Measured in Both Assays

In order to test whether the estimated exchange rates were ANT1-specific, we measured exchange rates at different ANT1 concentrations for both assays (Figure 4A). From a linear fit of the data sets, we obtained specific exchange rate values of k = 3.49 ± 0.41 mmol/min/g in the fluorescence assay and k = 2.90 ± 0.47 mmol/min/g in the radioactivity assay.

We further analyzed the specificity of both assays to ANT1, replacing ANT1 with UCP1, which binds but does not transport ATP and ADP. We measured exchange values of k_UCP1_ = 2.12 ± 0.57 mmol/min/g in the fluorescence assay and k_UCP1_ = 0.30 ± 0.10 mmol/min/g in the radioactivity assay (Figure 4B and Appendix A). Whereas the MgGr^TM^ assay measured the ADP/ATP ratio indirectly via free Mg^2+^, the amount of ^3^H-ATP was determined directly by liquid scintillation counting. Therefore, we assume that the alteration of the ADP/ATP ratio due to the presence of UCP1 in proteoliposomes may have resulted in a false-positive ADP/ATP exchange response in the fluorescence assay, which is absent in the radioactivity assay.

### 3.5. Inhibition of ANT1-Mediated ADP/ATP Exchange

To test the suitability of the fluorescence-based assay, we measured the ADP/ATP exchange with increasing concentrations of the ANT specific inhibitors, BA and CATR. The addition of BA and CATR decreased ADP/ATP exchange in a dose-dependent manner (Figure 5A). From an exponential function fit of the data, we obtained EC50 values of 55.6 ± 17.0 µM for BA and 32.5 ± 5.9 µM for CATR (Figure 5B), and maximum inhibition values of 91.1 ± 12.9% for BA and 83.1 ± 4.7% for CATR, respectively (Figure 5C).

## 4. Discussion

In the present study, we established a fluorescence-based method for proteoliposomes, which was developed initially for isolated mitochondria [7,16], and measured ANT1-dependent ADP/ATP exchange, which was inhibited by the ANT-specific inhibitors, BA and CATR. Although the radioactive assay measures the ATP amount in the proteoliposomes directly, its experimental use is unattractive due to the extensive precautions required to prevent radioactive contamination.

Other methods exist to continuously determine ADP/ATP exchange kinetics: (i) directly, using fluorescent derivatives of ATP or ADP [17] or (ii) indirectly, using luminescence of firefly luciferase [18], or fluorescence by NADP^+^ reduction [19,20]. However, their transfer to the model system is not as straightforward as the MgGr^TM^ fluorescence assay. The luminescence assay requires firefly luciferase, Mg^2+^ and oxygen for the conversion of ATP to AMP and pyrophosphate (PP_i_) [21], which alters the ADP/ATP ratio, increases AMP content and produces PP_i_, which is also supposed to interact with ANT activity [22]. Furthermore, the optimal working range is limited in terms of temperature and ionic strength; a variation in these conditions may decrease the sensitivity of measurements. Fluorescence measurements of NADP^+^ depend on glucose and enzymes, which are present in isolated mitochondria but have to be added in a model system. The use of fluorescent adenosine nucleotide derivatives suffers from substrate specificity of the ANT-mediated ADP/ATP exchange, since the full structures of the adenosine nucleotides are recognized [23]. Small variations in PN structure are supposed to significantly alter binding and transport of ATP and ADP [24,25,26,27,28,29], and further depend on the configuration of ANT [30].

The MgGr^TM^ based fluorescence method was primarily described for measurements in cell systems or isolated mitochondria. However, its precision is diminished by environmental stress, metabolic state and the presence of further mitochondrial carriers, such as the ATP-Mg/Pi carrier, which substantially alter the assay-relevant substrates. Here, we worked out the fluorescence technique for the model system of recombinant ANT1 reconstituted into proteoliposomes. This system benefits from clear predefined experimental conditions which allowed to calibrate fluorescence intensity in relation to Mg^2+^, ATP and ADP more precisely. Since ANT neither transports Mg^2+^ nor MgGr^TM^, the fluorescence intensity solely reflects alterations of the ADP/ATP ratio in the liposomes, and directly correlates with the ADP/ATP exchange activity of ANT.

A disadvantage of the model system is the low protein to lipid ratio of the reconstituted protein (<10 µg/mg lipid) compared to mitochondria, and the ratio of the transported substrate to bulk substrate is unfavorable. We compensated for the low signal-to-noise ratio by the presence of Mg^2+^ and MgGr^TM^ inside the (proteo-) liposomes. The presence of ATP inside the (proteo-) liposomes in the initial state is preferred since its exchange for ADP increases fluorescence, which is in contrast to a potential loss of intensity by photobleaching of MgGr^TM^ or leakage of Mg^2+^ through the membrane.

A further limitation originates from the kinetics of binding of Mg^2+^ to ADP and ATP. The amount of ATP and ADP must be set in order to significantly change the free Mg^2+^ concentration in liposomes upon their exchange. ATP and ADP concentrations below approximately 200 µM or above 3 mM (Figure 1C) will not significantly alter the free Mg^2+^ concentration and fluorescence intensity, and the ADP/ATP exchange cannot be further resolved by fluorescent dye.

Radioactivity assays are the gold standard to measure transport kinetics of membrane carriers with sufficient sensitivity. In order to verify the method and show the functionality of recombinant ANT1, we performed a standard radioactivity assay and obtained an exchange rate of 2.90 ± 0.47 mmol/min/g. The value was obtained at 24 °C in the absence of a transmembrane potential. In the literature, the measured exchange values of ANT vary significantly and strongly depend on the experimental conditions used, the tissue, the protein origin and the mitochondrial transmembrane potential. In isolated mitochondria, a strong temperature dependency of exchange kinetics was measured with rates ranging from 40 µmol/min/g at 10 °C, up to 550 µmol/min/g at 37 °C in rat liver, and from 100 µmol/min/g to 1.8 mmol/min/g in bovine heart [31]. Similar values were obtained in rat heart mitochondria by [32] and from isolated ANT from bovine heart mitochondria reconstituted into liposomes [10,33]. With the advent of yeast (y)ANTs (especially the yANT2 isoform) as a tool to investigate the transport function of ANT, the exchange rates were significantly raised by one order of magnitude to 7.5 mmol/min/g at 10 °C [11] and 65.7 mmol/min/g at room temperature [12].

Measurements performed with MgGr^TM^ in isolated rat mitochondria from different tissues produced ATP turnover rates from 5 s^−1^ in heart to 99 s^−1^ in liver, measured at 37 °C, and a transmembrane potential of −160 to −170 mV [7]. Our measured exchange rate of 3.49 ± 0.41 mmol/min/g corresponds to an ATP turnover rate of ≈ 1.75 s^−1^ and is substantially lower, but may be explained by the lower temperature and absence of a transmembrane potential.

Generally, fluorescent and radioactive assays have their advantages and disadvantages. The fluorescence assay is less expensive and allows for continuous measurements of the same sample, which decreases statistical variation. However, ANT-mediated exchange rates are measured indirectly via the free Mg^2+^ concentration as a response to alterations in the ADP/ATP ratio. Thus, a precise calibration of the system is mandatory. The radioactivity assay directly measures the ^3^H-ATP concentration in liposomes, but suffers from the increased effort to obtain a similar time course of ADP/ATP exchange. Furthermore, sample variation is increased as a further technical step is needed to remove external ^3^H-ATP at each time point.

Apart from substrate transport function, several members of the SLC25 family, including ANT [34,35], aspartate/glutamate carrier [36], dicarboxylate carrier [37], and uncoupling proteins [6], are suggested to have dual transport functions and are additionally involved in the fatty acid (FA)-mediated transport of protons. In this context, substrate transport reflects a major control for the protein’s functionality. For ANT1, the ease of the fluorescent assay allows for quick functionality control.

Of special interest is the possible use of the fluorescence assay to simultaneously measure FA-induced proton leak electrophysiologically and substrate transport fluorometrically. The parallel investigation will give new insights into the interplay between the two functions of ANT, exchanging ATP against ADP under coupled conditions, and transporting FA-mediated protons under uncoupled conditions.

## Figures and Tables

**Figure 1 biomolecules-10-00685-f001:**
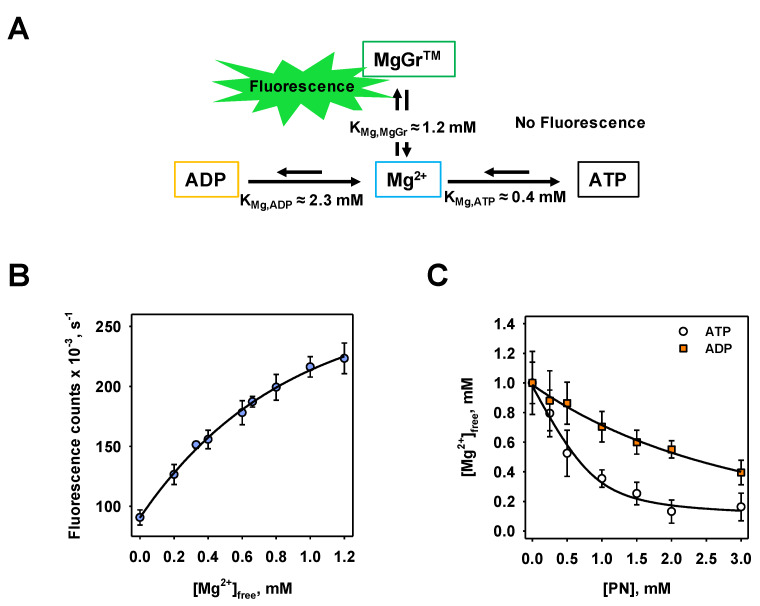
Working principles for ADP/ATP exchange measurements using MgGr^TM^ fluorescent dye. (**A**) A schema demonstrating the principle of the method. Mg^2+^ binds to MgGr^TM^, ATP and ADP with different affinities that results to the formation of fluorescent (Mg-MgGr^TM^) or non-fluorescent (Mg-ATP) complexes. (**B**) Dependence of fluorescence intensity of MgGr^TM^ on Mg^2+^ concentration. The line is the fit of an exponential rise to maximum function to the data. (**C**) Calculated concentration of free Mg^2+^ against purine nucleotides (PNs) ATP (open circles) and ADP (orange squares) concentrations. The initial concentration of Mg^2+^ was 1 mM. Lines derive from data fittings to the Michaelis–Menten function. In all measurements, the buffer contained 50 mM Na_2_SO_4_, 10 mM Tris, 10 mM MES and 0.6 mM EGTA at pH = 7.34 and T = 297 K. The concentration of MgGr^TM^ was 3 µM. Data are shown as the mean ± SD of at least three independent measurements.

**Figure 2 biomolecules-10-00685-f002:**
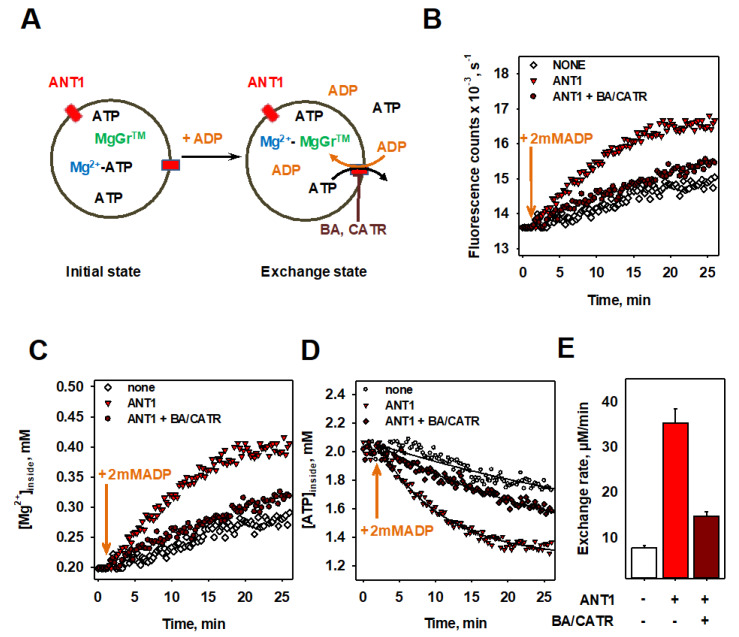
ADP/ATP exchange rate measurements using a fluorescence-based assay. (**A**) A scheme of the fluorescence-based assay of (proteo-)liposomes with internal Magnesium Green (MgGr)^TM^. The measurements started with the addition of ADP at time point t = 2 min (arrows). (**B**) Time dependence of the fluorescence intensity, (**C**) calculated [Mg^2+^], and (**D**) calculated [ATP] in the presence (red triangles) and absence (open circles) of ANT1, and in the presence of BA and CATR (red diamonds). (**E**) Maximal ADP/ATP exchange rate of liposomes (open bars) and ANT1 containing proteoliposomes in the absence (red bars) and presence (dark red bars) of ANT1 inhibitors (BA and CATR). Values were obtained as a fit parameter from (**D**). In all measurements, the buffer contained 50 mM Na_2_SO_4_, 10 mM Tris, 10 mM MES and 0.6 mM EGTA at pH = 7.34 and T = 297 K. Lipid membranes were made of DOPC, DOPE and CL (45:45:10 mol%). The lipid concentration was 1.0 mg/mL and the liposome diameter was approximately 100 nm. Concentrations of ATP, ADP, Mg^2+^ and MgGr^TM^ were 2 mM, 2 mM, 1 mM and 3 µM, respectively. The concentrations of ANT1 and inhibitors (BA, CATR) were 8.67 µg/mL and 100 µM, respectively. Data are shown as the mean ± SD of at least three independent measurements.

**Figure 3 biomolecules-10-00685-f003:**
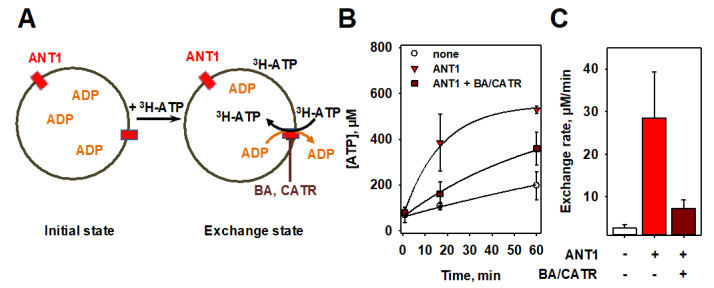
ADP/ATP exchange rate measurements of ANT1 using a radioactive assay. (**A**) Scheme of the ^3^H-ATP uptake assay. (**B**) Time course of the ATP concentration inside (proteo-)liposomes as determined by ^3^H-ATP uptake assay in the presence of ANT1 (triangles), ANT1 plus bongkrekic acid (BA) and carboxyatractyloside (CATR, squares), and in the absence of protein (circles). Lines are fits of an exponential rise to maximum function to the data. (**C**) Values of the ADP/ATP exchange rate as determined from the fit parameters in (**B**). In all measurements, the buffer contained 50 mM Na_2_SO_4_, 10 mM Tris, 10 mM MES, and 0.6 mM EGTA at pH = 7.34 and T = 297 K. Lipid membranes were made of 45:45:10 mol% of DOPC, DOPE and CL, respectively. The lipid concentration was 1.0 mg/mL and the liposome diameter was approximately 100 nm. The concentration of ANT1 was 8.67 µg/mL and the concentrations of ATP, ADP, BA and CATR were 2 mM, 2 mM, 100 µM and 100 µM, respectively. Data are shown as the mean ± SD of at least three independent measurements.

**Figure 4 biomolecules-10-00685-f004:**
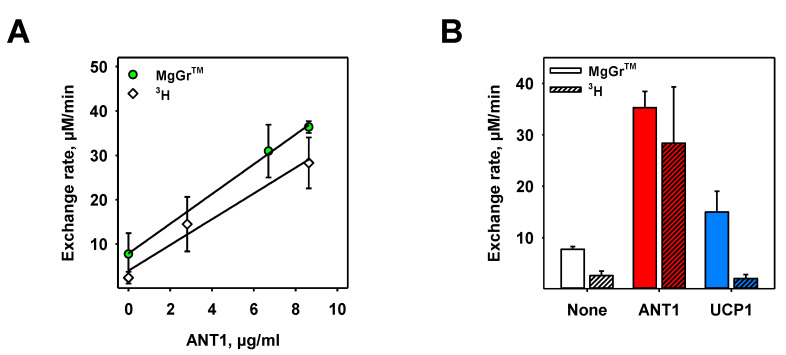
Determination of ANT1 specificity of ADP/ATP exchange. (**A**) ADP/ATP exchange rates at different ANT1 concentrations measured in the fluorescent (green circles) and radioactive (open diamonds) assay. Lines are a linear fit to the data. (**B**) Exchange rates measured in the absence of protein (open bars), and in the presence of ANT1 (red bars) and uncoupling protein 1 (UCP1, blue bars) in liposomes measured with Magnesium Green (MgGr)^TM^ (left bar of each data set) and ^3^H radioactivity (right bar of each data set). ANT1 and UCP1 were used at 8.67 µg/mL and 7.07 µg/mL, respectively. In all measurements, the buffer contained 50 mM Na_2_SO_4_, 10 mM Tris, 10 mM MES and 0.6 mM EGTA at pH = 7.34 and T = 297 K. Lipid membranes were made of 45:45:10 mol% of DOPC, DOPE and CL, respectively. The lipid concentration was 1.0 mg/mL and the liposome diameter was approximately 100 nm. Concentrations of ATP, ADP, Mg^2+^ and MgGr^TM^ were 2 mM, 2 mM, 1 mM and 3 µM, respectively. Data are shown as the mean ± SD of at least three independent measurements.

**Figure 5 biomolecules-10-00685-f005:**
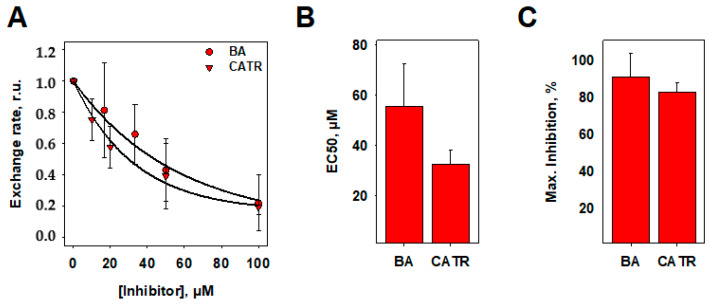
Dose-dependence of ANT1-mediated ADP/ATP exchange in the presence of specific inhibitors (CATR and BA). (**A**) Dose-response curves of ANT1-mediated ADP/ATP exchange rates measured with a fluorescence assay in the presence of bongkrekic acid (BA, circles) and carboxyatractyloside (CATR, triangles) normalized to the full, uninhibited rate. Lines are fits of an exponential function to the data. (**B**) EC_50_ and (**C**) maximum inhibition by BA and CATR obtained as fit parameters from (**A**). In all measurements, the buffer contained 50 mM Na_2_SO_4_, 10 mM Tris, 10 mM MES and 0.6 mM EGTA at pH = 7.34 and T = 297 K. Lipid membranes were made of 45:45:10 mol% of DOPC, DOPE and CL, respectively. The lipid concentration was 1.0 mg/mL and the liposome diameter was approximately 100 nm. The concentration of ANT1 was 6.70 µg/mL, and the concentrations of ATP, ADP, Mg^2+^, and Magnesium Green (MgGr)^TM^ were 2 mM, 2 mM, 1 mM and 3 µM, respectively. Data are shown as the mean ± SD of at least three independent measurements.

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
