# Peer review of "A Fluorescence-Based Method to Measure ADP/ATP Exchange of Recombinant Adenine Nucleotide Translocase in Liposomes"

_biomolecules, 2020, doi:10.3390/biom10050685_

Round 1

Reviewer 1 Report

The primary objective of this manuscript is to evaluate a fluorescence based method to measure the transport of ATP and ADP through adenine nucleotide translocase in mitochondria.  The authors instead use proteoliposomes as a model system for mitochondria and their experiments.  The exchange rate measurements correlate between the fluorescence-based assay developed by the authors and the traditional measurements performed by radioactive assays.  It would have been nice for the authors to attempt to use their system to measure ATP transport in real mitochondria, but the discussion addresses possible limitations.  Nevertheless, this is a worthy technique for publication.

Author Response

The primary objective of this manuscript is to evaluate a fluorescence based method to measure the transport of ATP and ADP through adenine nucleotide translocase in mitochondria.  The authors instead use proteoliposomes as a model system for mitochondria and their experiments.  The exchange rate measurements correlate between the fluorescence-based assay developed by the authors and the traditional measurements performed by radioactive assays.  It would have been nice for the authors to attempt to use their system to measure ATP transport in real mitochondria, but the discussion addresses possible limitations.  Nevertheless, this is a worthy technique for publication.

We thank the reviewer for the positive feedback. Since we deemed to develop the method for the use in proteoliposomes, the measurements in mitochondria would be out of scope for this study. Moreover, the similar fluorescence measurements in isolated mitochondria were already described.

Reviewer 2 Report

This submission demonstrated a fluorescence- based method using Magnesium Green (MgGrTM), a Mg2+-sensitive dye suitable for measurement in liposomes. It seems useful and helpful in the research. I like to give the following comments.

  1. As mentioned in the introduction, Mg2+ is a crucial co-factor and second messenger in cells. Therefore, specificity to target for assay is extremely important. Please introduce it in detail.
  2. In Figure 1C, what is meaning of PN that was not indicated in the legends.
  3. In Figure 2A, uptake of Magnesium Green (MgGr)TM into ANTI liposome during initial state may have limitation(s). Please concern it carefully.
  4. Concentration of ADP in Figure 2 is also important.
  5. In Figure 3, legends show errors that need o revise carefully.
  6. The transmembrane potential seems also important in the assay, particularly the mitochondrial samples.
  7. In the future, the commercial kit may cost lower than radioisotope assay. It is another advantage that was not mentioned in the context.

Author Response

This submission demonstrated a fluorescence- based method using Magnesium Green (MgGrTM), a Mg2+-sensitive dye suitable for measurement in liposomes. It seems useful and helpful in the research.

We thank the reviewer for this comment.

I like to give the following comments.

  1. As mentioned in the introduction, Mg2+is a crucial co-factor and second messenger in cells. Therefore, specificity to target for assay is extremely important. Please introduce it in detail.

We have added the following underlined expression to the sentence in the introduction, page 2, lines 43 to 48 to outline the unspecific effect of Mg2+ in cells and mitochondria in the assay:

Magnesium is the second most abundant cation in the cell and is involved in more than 300 cellular and mitochondrial enzymatic reactions, including metabolism of nucleic acids, lipids and proteins (DOI: https://doi.org/10.2741/1223). In particular, Mg2+ participates in reactions involving the formation and use of ATP. Therefore, the assay specificity in living mitochondria is difficult to achieve and the estimation of the free Mg2+ concentration in mitochondria is delicate and may rapidly change during measurements.

  1. In Figure 1C, what is meaning of PN that was not indicated in the legends.

Thank you for noticing, we have added the underlined expression to the legend for figure 1 on page 5, lines 170 to 172:

Calculated concentration of free Mg2+ against purine nucleotides (PNs) ATP (open circles) and ADP (orange squares) concentrations.

  1. In Figure 2A, uptake of Magnesium Green (MgGr)TM into ANTI liposome during initial state may have limitation(s). Please concern it carefully.

We are aware that Magnesium Green is available as cell-permeable and cell-impermeable fluorescent dye. We used the cell-impermeable MgGrTM  (ThermoFisher, catalog number M3733). At the beginning of the experiment we removed the extra-liposomal fluorescent dye by size exclusion chromatography to ensure that it is only present inside the liposomes.

We added the full name of the dye and catalog number to the material and methods, part 2.1 chemicals on page 2, lines 70 to 71.

  1. Concentration of ADP in Figure 2 is also important.

We changed the expression “+ ADP” to the expression “+ 2 mM ADP” in the figures 2,  B-D  on page 6 to indicate the concentration of added ADP, accordingly.

  1. In Figure 3, legends show errors that need to revise carefully.

We are thankful to the reviewer to notice the mistake. We have substituted the sentence on page 6, lines 221 to 222 by the following underlined expression:

The concentration of ANT1 was 8.67 µg/mL and the concentrations of ATP, ADP, BA and CATR were 2 mM, 2 mM, 100 µM and 100 µM, respectively.

We added the abbreviations BA and CATR in lines 215 and 216, page 6, respectively.

  1. The transmembrane potential seems also important in the assay, particularly the mitochondrial samples.

We added the underlined expression to the sentence in the discussion, page 9, lines 318 to 321:

In the literature, the measured exchange values of ANT vary significantly and strongly depend on the experimental conditions used - the tissue, the protein origin and the mitochondrial transmembrane potential.

  1. In the future, the commercial kit may cost lower than radioisotope assay. It is another advantage that was not mentioned in the context.

We thank the author to mention the cost relation. Indeed, the fluorescence dye is cheaper compared to the radionucleotides. Respectively, we introduced the following underlined expression in the discussion to the sentence on page 9, lines 333 to 335:

The fluorescence assay is less expensive and allows for continuous measurements of the same sample, which decreases statistical variation.